# The Levels of Markers of Muscle Damage, Inflammation, and Heat Shock Proteins in Judokas and the Extent of Their Changes during a Special Performance Test at Different Ambient Temperatures

Tomasz Pałka [1], Tadeusz Ambroży [2], Ewa Sadowska-Krępa [1], Łukasz Rydzik [2,*], Szczepan Wiecha [3], Marcin Maciejczyk [1], Peter Kacúr [4], Piotr Michał Koteja [1], Bibiana Vadašová [5], Kazimierz Witkowski [6] and Łukasz Tota [1]

[1] Department of Physiology and Biochemistry, Faculty of Physical Education and Sport, University of Physical Education in Kraków, 31-571 Kraków, Poland; tomasz.palka@awf.krakow.pl (T.P.); e.sadowska-krepa@awf.katowice.pl (E.S.-K.); lukasz.tota@awf.krakow.pl (Ł.T.)
[2] Institute of Sports Sciences, University of Physical Education in Kraków, 31-571 Kraków, Poland; tadek@ambrozy.pl
[3] Faculty in Biala Podlaska, Department of Physical Education and Health, Jozef Pilsudski University of Physical Education in Warsaw, 00-968 Biala Podlaska, Poland
[4] Department of Sports Educology and Humanistics, Faculty of Sports, University of Presov, 080 01 Presov, Slovakia; peter.kacur@unipo.sk
[5] Department of Sports Kinanthropology, Faculty of Sports, University of Presov, 080 01 Presov, Slovakia; bibiana.vadasova@unipo.sk
[6] Faculty of Physical Education and Sports, University of Physical Education in Wrocław, 51-612 Wrocław, Poland; kazimierz.witkowski@awf.wroc.pl
* Correspondence: lukasz.rydzik@awf.krakow.pl

**Abstract:** Background: Athletes in combat sports, such as judo, often experience muscle cell damage due to physical and metabolic stress. This study investigates the impact of anaerobic interval exercises involving both upper and lower limbs at different temperatures on physiological indicators. Methods: Fifteen judokas, with an average age of $20.7 \pm 2.0$ years, participated in the study. They had an average body height of $178 \pm 6.3$ cm, body mass of $76.3 \pm 12.6$ kg, VO$_2$max of $43.2 \pm 7.8$ mL·kg$^{-1}$, and peak power of 12.1 W·kg$^{-1}$. The main experiment involved performing five sequences of pulsating exercise on a cycle ergometer for both upper and lower limbs. This was conducted in a thermoclimatic chamber set at temperatures of $21 \pm 0.5$ °C and $31 \pm 0.5$ °C with a relative humidity of $50 \pm 5\%$. The sequences alternated pulsations of varying durations and loads between the upper and lower limbs, with a 15 min break following each sequence. Within each sequence, participants underwent four anaerobic limb tests. Biochemical markers, including interleukin-1β (IL-1β), interleukin-6 (IL-6), and heat shock protein 70 (HSP70), were measured using the enzyme-linked immunosorbent assay (ELISA) method before and after exercise, and again at 1, 24, and 48 h post-exercise. Muscle cell damage was evaluated based on lactate dehydrogenase (LDH) activity and myoglobin (Mb) concentration. Results: Both temperature conditions elicited physiological and biochemical responses. Positive correlations were observed between white blood cell count (WBC) and LDH concentration at 21 °C, as well as between WBC and IL-6 at 21 °C. At 31 °C, correlations were seen between WBC and myoglobin, and WBC and LDH. Conclusions: Muscle cell damage was evident under both conditions, as indicated by increased myoglobin levels. These findings offer insights into training strategies and underscore the physiological responses observed in combat sports athletes.

**Keywords:** judo; physical capacity; temperature; LDH; MB; IL

## 1. Introduction

The metabolic processes during judo training or tournaments predominantly depend on anaerobic mechanisms. These processes can lead to substantial muscle cell damage due to mechanical and metabolic stress, especially in high-temperature environments [1,2]. Such stress triggers an inflammatory state, marked by cytokine-induced structural and metabolic alterations in the affected areas [3,4]. At the site of injury, neutrophils and their associated pro-inflammatory cytokines, as well as reactive oxygen and nitrogen species, become evident [5]. Additionally, muscle proteins, such as myoglobin (Mb), are released into the bloodstream [5]. Together with neuroendocrine factors, muscle damage significantly influences the immune response to exercise [3,6].

Judo athletes in competitions face multiple rounds, each roughly spanning 5 min [7]. The structure of these competitions includes intermittent sequences of combat and rest periods, amounting to approximately 460 s (7 min 40 s) in total [7]. Competing in higher ambient temperatures strains the body's systems more than performing at room temperature, potentially because of shifts in plasma volume indicating dehydration [8].

Muscle damage from exercise prompts a rise in pro-inflammatory interleukin production, especially interleukin-6 (IL-6) and interleukin-1β (IL-1β), in conjunction with cytokine-driven neutrophil and macrophage reactions [3,9]. These cytokines also stimulate the production of heat shock proteins (HSPs) that possess anti-inflammatory capabilities [10,11]. The expression of IL-6 correlates with the "energy deficit" encountered during physical exertion, prompting the release of energy compounds [12]. Moreover, skeletal muscle injuries enhance the secretion of pro-inflammatory cytokines such as IL-1β [13–15].

Cytokines also promote the synthesis of acute-phase proteins in the liver. IL-6 activates T lymphocytes, modulates B lymphocyte differentiation, and amplifies HSP production [16]. A rise in internal body temperature by 1 °C due to physical exercise instigates catecholamine and cytokine release, which in turn affects leukocyte circulation [17]. The liver's increased amino acid uptake for HSP synthesis might lower blood glutamine levels, subsequently affecting immune functionality [18].

There is a paucity of comprehensive data about the inflammation triggered by high-intensity, acyclic workouts, particularly in hot environments [19]. Heat shock proteins (HSPs) serve as chaperones, shielding proteins from atypical alterations under stressful conditions [19]. The synthesis of HSPs can occur both under stress and in standard physiological states [20]. HSP70 also influences immune responses by affecting the major histocompatibility complex (MHC), with its expression induced by multiple factors [21]. Elevated ambient temperatures notably boost HSP70 levels [20,22].

Physical activity-related cell injuries instigate an inflammatory response, marked by an uptick in acute-phase proteins, predominantly propelled by IL-6 [20]. Exercise intensity, duration, engaged muscle groups, and activated fiber types all affect HSP70 and HSP72 concentrations [19]. HSP70 helps mitigate micro-damage resulting from anaerobic exercises, and individual variability can influence its expression [23]. Acclimating to heat can modify the immune system, leading to increased concentrations of anti-inflammatory interleukin (IL-10) and HSP72 [24]. HSP levels might serve as indicators for heat stress vulnerability [19].

This research addresses an existing knowledge gap regarding high-intensity, acyclic exercises in hotter climates, focusing on their impact on physiological and biochemical markers [19]. The insights gathered could inform optimized training regimens for athletes [25]. The study's primary objective is to ascertain if interval anaerobic workouts targeting both upper and lower limbs, executed in varied ambient temperatures, have a consistent effect on muscle damage indicators and blood-borne biochemical markers.

## 2. Materials and Methods

A group of 15 elite male professional judo athletes, representing the country's best, were chosen for the study from a pool of 20 candidates. These athletes all had up-to-date medical examinations. The selection criteria emphasized chronological age, training

experience, and competitive level—every athlete had previously ranked within the top 5 in national competitions (Table 1). Given the elite caliber of the participants, the sample size was determined using G*Power. Out of the initial group, 10 athletes completed the full testing cycle. The study received approval from the Bioethics Committee of the Krakow Regional Medical Chamber (No. 102/KBL/OIL/2011) and was funded through statutory research funds (7/BS/IFC/2011). All male participants were briefed about the study's objectives, methodology, potential side effects, and their right to opt out at any point, in line with the Helsinki Declaration. They provided written consent to partake. The entire study was executed under medical oversight.

**Table 1.** Inclusion and exclusion criteria.

| 20 Professional Judo Competitors | |
|---|---|
| Included: 15 | Excluded: 5 |
| Inclusion criteria | Exclusion criteria |
| Age > 18 years | Age < 18 years |
| Good health status | Diseases and injuries |
| High sports skill level | Low sports skill level |
| Training experience > 10 years | Too short training experience |
| At least 5th place in national competitions | No success in national competitions |
| Examination started by 15 participants | |
| Full scope of tests completed by 10 participants | Full scope of tests not completed by 5 participants |

Endurance tests for the judokas took place during the competition period, housed within a thermoclimatic chamber and an air-conditioned laboratory at the Department of Physiology and Biochemistry, Academy of Physical Education in Krakow. To account for circadian rhythms, all tests were scheduled in the morning, ensuring they began at least 2 h post a light meal. The testing for judo athletes was bifurcated into preliminary (Stages I, II, and III) and main phases (Stages IV and V).

*2.1. Participant Characteristics*

Of the athletes who entered the study, ten completed the entire research program. Their average age was $20.7 \pm 2.0$ years. These men had a mean body height (BH) of $178 \pm 6.3$ cm, body mass (BM) of $76.3 \pm 12.6$ kg, lean body mass (LBM) of $64 \pm 10.4$ kg, and a body surface area to body mass ratio (BSA·BM$^{-1}$) of $0.0249 \pm 0.0016$ cm$^2$·kg$^{-1}$. In terms of aerobic capacity, their VO$_2$max was $43.23 \pm 7.8$ mL·kg$^{-1}$ for the lower limb test, and $3.3 \pm 0.63$ L·min$^{-1}$ in absolute values. Their anaerobic capacity (RPP) averaged at $12.12$ W·kg$^{-1}$.

*2.2. Preliminary Research*

STAGE I

In the first stage of the study, basic biometric measurements were taken, including height (BH) and body mass (BM). These measurements were utilized to calculate the Quetelet II index (BMI) as per Dubois. The lean body mass (LBM) was estimated, along with the body surface area (BSA) and the ratio of body surface area to body mass (BSA·BM$^{-1}$). Additionally, for diagnostic objectives, athletes' blood pressure (BP) and heart rate (HR) were measured.

STAGE II

Male participants undertook exercise tests to gauge both anaerobic and aerobic endurance of the lower limbs (LL).

Stage III

(seven days later): The same tests were administered, but this time focused on the upper limbs (UL).

For both stages, biometric measurements preceded the exercise tests. The Wingate test [26] was employed to measure anaerobic endurance, both for the LL and UL. Before the main test, a 5 min warm-up on a cycle ergometer was conducted with an intensity tailored to 50% of the participant's $VO_2$max, set at a frequency of 60 rpm. This warm-up included three 5 s maximum accelerations at the 2, 4, and 5 min marks. Two minutes post-warm-up, the participants executed a 30 s maximum physical exertion. For the LL, an external resistance equal to 8.3% of the subject's body weight was applied, and for UL, the resistance was set at 4.5% of their body mass [27]. Key Wingate test parameters were observed and analyzed. Using the derived data, anaerobic endurance, encompassing both phosphagen and glycolytic elements, was evaluated. Blood samples were taken from the earlobe before and three minutes post-anaerobic effort to enzymatically determine the lactate concentration.

A minimum of two hours after the Wingate Test, participants underwent an aerobic capacity evaluation. This involved an incremental test "to subjective exhaustion", a standard procedure at the Department of Physiology and Biochemistry at the Academy of Physical Education in Krakow. The conditions for the test were a room temperature maintained at $21 \pm 0.5\,°C$ with a relative humidity of $40 \pm 3\%$. Each test began with a two-minute warm-up on a cycle ergometer, maintaining 60 RPM. The initial intensity was set at 110 W for LL and 60 W for UL. After the warm-up, for the LL, power output increased by 20 W and for the UL, it increased by 12 W every two minutes. The participants persisted in the exercise until they subjectively felt they could not maintain the designated pedaling pace. Blood samples were collected from the earlobe both before and after this test to determine lactate concentration (LA).

*2.3. Main Study*

Stage IV and V

Stage IV: Half of the judo participants executed 5 restrictive pulsatile efforts on the leg and arm cycle ergometer inside a thermoclimatic chamber, set at a temperature of $21 \pm 0.5\,°C$.

Stage V: The other half performed the same exercises, but at a higher temperature of $31 \pm 0.5\,°C$.

For both stages, the relative humidity was maintained at $50\% \pm 5\%$.

After a recovery period of seven days, essential for offsetting the effects of physical exertion, the participants switched conditions and revisited Stages IV and V. Specifically, those who initially worked out at $21 \pm 0.5\,°C$ now exercised at $31 \pm 0.5\,°C$ and vice versa.

These exercises were not your standard interval pulsatile efforts. They were uniquely timed and alternated between loading the upper and lower limbs during anaerobic bursts of physical activity. These anaerobic sessions were punctuated with 15 min rest breaks (as depicted in Figure 1).

A single series of pulsatile exercise was conducted based on the scheme outlined below and was repeated five times (as detailed in Table 2). The complete duration of the experiment, which included the five series of pulsatile exercise sequences along with four 15 min breaks interspersed between them, totaled 96 min and 20 s. In competitive settings, the best judo athletes participate in 4 to 6 matches. Each match lasts 7 min and 40 s. In our study, considering the duration of an individual match and the typical tournament format, we designed a sequence of efforts that most closely mirrored the competitive structure in judo.

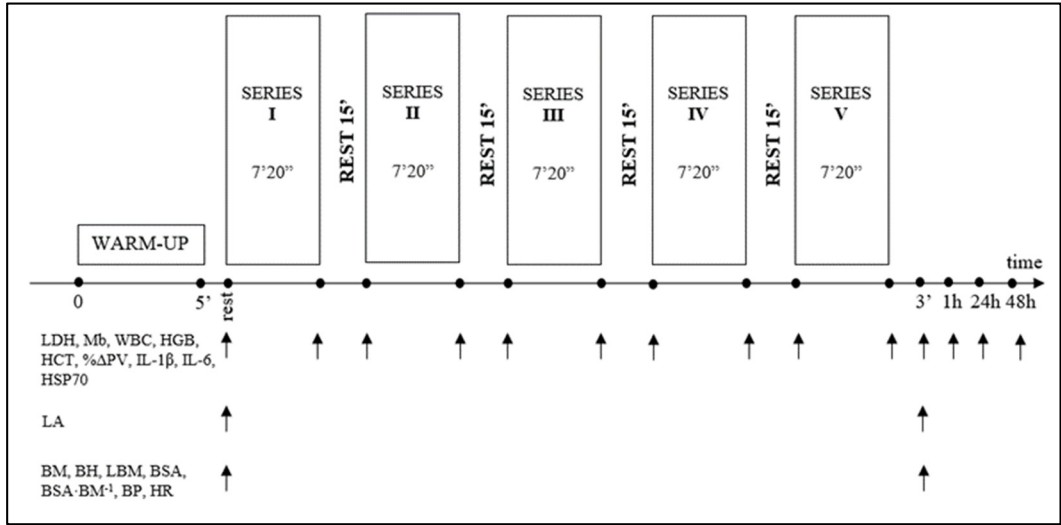

**Figure 1.** Diagram of the study with measurement points.

**Table 2.** Characteristics of a single series of alternate pulsating efforts.

| LL | I | UL | I | LL | I | UL | I | LL | I | UL | I | LL | I | UL | I |
|------|-------|------|-------|------|-------|------|-------|------|-------|------|-------|------|-------|------|-------|
| 15 s. | 30 s. | 15 s. | 30 s. | 30 s. | 60 s. | 30 s. | 60 s. | 20 s. | 45 s. | 20 s. | 45 s. | 15 s. | 30 s. | 15 s. | 30 s. |

UL—upper limbs, LL—lower limbs, I—interval.

Before the pulsatile exercise at both temperatures, participants underwent a 30 min acclimatization to the thermal conditions. This was followed by a 5 min warm-up at an individually tailored load of $50 \pm 1\%$ VO$_2$max. The warm-up maintained a frequency of 60 rev·min$^{-1}$ and incorporated three 5 s maximal accelerations at the 2nd, 4th, and 5th minutes. During each sequence of the pulsatile test—modified from the Wingate test for both lower (LL) and upper (UL) limbs—fundamental Wingate test metrics were recorded. The load remained consistent for each sequence: 8.3% of body mass for LL and 4.5% for UL. Moreover, before and after the five series of the pulsatile test, biochemical markers were identified from blood samples taken at intervals of 1, 24, and 48 h. These markers included interleukin-1β (IL-1β), interleukin-6 (IL-6), and heat shock proteins (HSP-70). To gauge muscle cell damage, metrics like lactate dehydrogenase (LDH) activity, myoglobin (Mb) concentration, leukocyte (WBC) count, hemoglobin (HGB) concentration, hematocrit (HCT) count, and changes in plasma volume (%ΔPV) were considered. The pulse test interval was structured to mimic the pattern of judo tournament matches, a design derived from an analysis of actual matches. This approach was inspired by related studies in the literature [28,29].

The aim of the chosen exercise protocol encompassed the following:

Sporting Bout Simulation: Testing in a thermoclimatic chamber ensures consistent control over temperature and humidity—critical variables during strenuous activity. The contrast in temperatures between the two stages (21 °C and 31 °C) facilitates an understanding of physiological responses to fluctuating thermal conditions one might face in a tournament [30].

Pulsed Interval Training: This exercise protocol, characterized by intermittent increase and decrease in load, mirrors the physical exertion witnessed in a judoka's tournament or training [31].

Diverse Limb Loading: The protocol factors in the varied loading of both upper and lower limbs are significant given the movement dynamics intrinsic to judokas. Training both areas simultaneously can bolster strength, endurance, and coordination [32].

Rest Breaks: The 15 min gaps offer athletes recovery time before the next sequence. This equilibrium between exertion and rest is crucial for efficient training and injury prevention. The workout intervals were tailored to reflect a judo match's timing [33].

The entire exercise protocol was structured to fine-tune the judokas' training to their sport's nuances, factoring in physical strain, varying thermal conditions, and inter-session recovery. Evaluating such a protocol can shed light on how these elements impact a judoka's conditioning and preparation. Furthermore, this can inform the development of advanced, effective training methodologies in subsequent years.

### 2.4. Environmental Monitoring

1. Temperature and Humidity: Monitored using a Harvia thermohygrometer (Muurame, Finland) and an Ellab electrothermometer (Hillerød, Denmark). The accuracies of these devices are $\pm 0.5\ ^\circ$C and $\pm 3\%$, respectively.
2. Air Movement: Measured with a Hilla catheter thermometer, utilizing the simplified Weiss formula for minor air movements below $1\ \mathrm{m \cdot s^{-1}}$.

### 2.5. Anthropometric Measurements

1. Height (BH): Measured with a Martin anthropometer (Charlotte, NC, USA) with an accuracy of 0.5 cm.
2. Body Mass (BM): The Sartorius F 1505-DZA electronic scale (Göttingen, Germany) was used, accurate to 1 g.

The collected BH and BM data facilitated the calculation of the Quetelet II index (BMI). Established norms for the group were 18.5–24.99, with values below 18.5 categorized as underweight and above 25 as overweight. Body fat percentage (PBF), body fat mass (MBF), and lean body mass (LBM) were analyzed via an eight-electrode bioelectrical impedance body composition analyzer (JAWON MEDICAL IOI-353, Seoul, Korea).

### 2.6. Blood Sampling and Analysis

All blood collection adhered to standard practices by a certified laboratory diagnostician.

1. Lactate Measurements: Blood was drawn from the earlobe into a 10 μL heparinized capillary. Lactate concentrations were determined enzymatically using a Mini-photometer plus DR Lange, type LP-20 (Dr Lange, Germany).
2. Hematological Measurements: 5 mL of venous blood was collected from the elbow crease into an EDTA tube. Leukocytes (WBC), hematocrit (HCT), and hemoglobin concentration (HBH) were determined using a Vet-Analyzer Hematology HA-22/20/ (CLINDIAG SYSTEMS, Ninove, Belgium).
3. Biochemical Measurements: Blood samples were taken before the pulsatile tests, one hour post-test, and at 24 and 48-h intervals. Blood was divided into tubes with a coagulation activator and EDTA, then centrifuged at 2000 rpm for 15 min in an MPW 351R Med. Instruments Polska centrifuge. Post-centrifugation, serum and plasma were isolated and frozen at $-70\ ^\circ$C (Artico ULF 390 ChRL freezer, Esbjerg, Denmark) for later analysis.

### 2.7. ELISA-Based Blood Marker Measurements

1. ELISA (Enzyme-Linked Immunosorbent Assay) is an acclaimed method for pinpointing specific proteins in biological specimens, leveraging monoclonal or polyclonal antibodies conjugated with a designated enzyme.

### 2.8. Muscle Damage Markers

1. Myoglobin: DRG Instruments GmbH, Marburg, Germany; sensitivity: 5 ng/mL
2. Lactate dehydrogenase (LDH): Wuhan Eiulb Science Co., Ltd., Wuhan, China; sensitivity: 2.8 U/mL

*2.9. Selected Cytokines*

1. Interleukin-6: DRG Instruments GmbH, Marburg, Germany; sensitivity: 2 pg/mL
2. Interleukin-1β: DRG Instruments GmbH, Marburg, Germany; sensitivity: 0.35 pg/mL

*2.10. Others*

1. Heat shock protein 70 (HSP-70): Wuhan Eiulb Science Co., Ltd., Wuhan, China; sensitivity: 0.039 ng/mL.
2. Plasma Volume Calculation.
3. Changes in plasma volume (%ΔPV) were ascertained using a formula adapted from Dill and Costill, further revised by Harrison et al. Post-exercise biochemical marker concentrations were adjusted for plasma volume changes employing the Kraemer and Brown method.
4. Hematocrit (HCT%) Measurement: Deployed the micro-method with a Unipan MPW-212 centrifuge.
5. Hemoglobin Concentration Determination: Utilized the Drabkin technique involving a spectrophotometer.
6. Physiological Measurements.
7. Heart Rate (HR): Monitored using a Polar 610S cardiomonitor.
8. Dehydration Degree: Evaluated from body mass measurements and urine volume.
9. Respiratory Exchange During Exercise: Assessed using Ergospirotest apparatus.
10. Graded Exercise Test: Executed on specific ergometers, varying for lower and upper limbs.
11. Pulsatile Anaerobic Tests: Conducted in a thermoclimatic chamber at specific temperatures.

*2.11. Statistical Methods*

Data analysis was executed using Statistica 9.0 for Windows. The study incorporated various statistical tests and computations such as multivariate analysis of variance, the Shapiro–Wilk test, the post-hoc Tukey test, and the Pearson correlation coefficient. Sample size determination was done via G*Power 3.1.9.4 software.

## 3. Results

The study involved judokas aged $20.65 \pm 2.03$ years who had been involved in sports for $10.36 \pm 1.5$ years. Prior to the main part of the study, the athletes performed a Wingate test to determine their anaerobic capacity level. Considering the specific physical demands of judo training and competition, a lower limb anaerobic test (LL) was performed in stage II of the research protocol, followed by an upper limb test (UL) in stage III. Aerobic endurance was assessed in the same stages using a graded exercise test (DE) and maximal oxygen uptake ($VO_2max$).

Preliminary study results showed that in the lower limb Wingate test, relative peak power (RPP) was $12.12 \pm 0.87$ W·kg$^{-1}$, while in the upper limb test, it was $7 \pm 0.56$ W·kg$^{-1}$, which was 42.2% lower. Total work (TW) in the lower limb test was $21.85 \pm 4.26$ kJ, while in the upper limb test, it was $13.36 \pm 2.5$ kJ. The difference was 38.6% (Table 3).

In both versions of the test, the resting and post-exercise values of lactate concentration were recorded. From the numerical data presented in Table 4, it can be concluded that the anaerobic physical effort performed with both the upper and lower limbs led to statistically significant changes ($p < 0.05$) in blood lactate concentration.

**Table 3.** Statistical characteristics of the Wingate test results performed with lower limbs (LL) and upper limbs (UL) in judo athletes.

| Index | LL $\bar{x}$ SD | UL $\bar{x}$ SD |
|---|---|---|
| PP (W) | $933.00 \pm 216.13$ | $536.09 \pm 107.73$ |
| RPP (W·kg$^{-1}$) | $12.12 \pm 0.87$ | $7.00 \pm 0.56$ |
| TW (kJ) | $21.85 \pm 4.26$ | $13.36 \pm 2.50$ |
| TW (J·kg$^{-1}$) | $285.27 \pm 17.73$ | $173.55 \pm 13.50$ |

LL—lower limbs, UL—upper limbs, TW—total works, PP—peak power, RPR—relative peak power, $\bar{x}$—mean, SD—standard deviation.

**Table 4.** Lactate concentration in blood collected before and after the Wingate test performed with the lower limbs (LL) and upper limbs (UL).

| Index | LL | | UL | |
|---|---|---|---|---|
| | Before | After | Before | After |
| La [mmol·L$^{-1}$] | $1.92 \pm 0.31$ | $14.11 \pm 1.37$ | $1.88 \pm 0.27$ | $13.05 \pm 1.16$ |
| $p<$ | <0.05 | | <0.05 | |

LL—lower limbs, UL—upper limbs, La—lactate, $p$—significance.

The aerobic endurance of the male participants was determined by the duration of the graded exercise test (DE). The numerical data in Table 3 indicate that in the lower limb version of the test, the duration of physical work was on average 2.68 min, or 15%, longer than in the upper limb version. The mean maximal oxygen uptake (VO$_2$max), considered the main indicator of aerobic capacity, reached $43.23 \pm 7.79$ mL·kg$^{-1}$·min$^{-1}$ in the lower limb test, and $37.19 \pm 5.26$ mL·kg$^{-1}$·min$^{-1}$ in the upper limb test. The difference was 13.97%. The mean maximal heart rate (HRmax) recorded in the graded exercise test with lower limbs was $185 \pm 8.19$ bpm, while in the upper limb test it was $183 \pm 8.43$ bpm (Table 5).

**Table 5.** Physiological and biochemical indices in the graded exercise test with lower (LL) and upper (UL) limbs.

| Index | LL $\bar{x}$ SD | UL $\bar{x}$ SD |
|---|---|---|
| VO$_2$max [mL·kg$^{-1}$, L·min$^{-1}$] | $43.23 \pm 7.79$ | $37.19 \pm 5.26$ |
| HRmax [h·min$^{-1}$] | $185 \pm 8.19$ | $183 \pm 8.43$ |

LL—lower limbs, UL—upper limbs, HR—heart rate, $\bar{x}$—mean, SD—standard deviation.

In both versions of the test, aerobic physical exercise performed with both the upper and lower limbs led to statistically significant changes in blood lactate concentration ($p < 0.05$). In the test performed with the lower limbs, the increase in blood lactate concentration ($\Delta$La) was on average 10.98 mmol·L$^{-1}$, while in the test performed with the upper limbs it was 8.34 mmol·L$^{-1}$ (Table 6).

**Table 6.** Blood lactate concentration measured before and after the graded exercise test performed with the lower limbs (LL) and upper limbs (UL).

| Index | LL | | UL | |
|---|---|---|---|---|
| | **Before** | **After** | **Before** | **After** |
| La [mmol·L$^{-1}$] | $1.97 \pm 0.20$ | $12.95 \pm 1.80$ | $1.87 \pm 23$ | $10.21 \pm 2.09$ |
| *p<* | <0.05 | | <0.05 | |

LL—lower limbs, UL—upper limbs, La—lactate, *p*—significance.

### 3.1. Results of Main Measurements—(Stage IV and V)

Selected physiological indicators recorded before and after pulsating physical exercise at ambient temperatures of 21 and 31 °C (Table 7).

**Table 7.** Mean and standard deviation of body mass (BM kg) in comparison to temperatures in individual measurements before and after a series of pulsating exercise.

| Temperature | Measurement | $\bar{x}$ SD |
|---|---|---|
| 21 °C | Before | $74.68 \pm 11.32$ |
| | After | $73.70 \pm 10.99$ # |
| 31 °C | Before | $74.87 \pm 11.29$ |
| | After | $73.31$ * $\pm 11.12$ |

#—significant differences at $p < 0.05$ at 21 °C—differences between measurements, *—significant differences at $p < 0.05$ at 31 °C—differences between measurements.

As a result of dehydration following a series of LL and UL exercises at temperatures of 21 °C and 31 °C, a statistically significant reduction in body mass was observed ($p < 0.05$). After exercise at 21 °C, body mass decreased by 0.980 kg, and at 31 °C, by 1.560 kg. There was no statistically significant difference between the BM values recorded before and after the exercise series at 21 °C and 31 °C.

### 3.2. Changes in Hematological Indicators of Blood after Pulsatile Exercise at Different Ambient Temperatures

Following the anaerobic pulsatile physical exercise series at ambient temperatures of 21 °C and 31 °C, leukocytosis was observed. Statistically significant differences ($p < 0.05$) in the number of white blood cells (WBC) were observed in blood samples taken at rest and at 1 h after the end of exercise. Statistically significant differences were also found between the number of WBC at 1 h after exercise and the number of WBC at 24 and 48 h after exercise. The highest post-exercise values were recorded in the first hour after the end of pulsatile exercise at 31 °C. These values were higher by 0.73 $10^3/\text{mm}^3$ than at 21 °C. Pearson's linear correlation results showed a positive relationship between the increase in WBC and Mb concentration in the first hour after physical exercise at 21 °C (r = 0.65; $p < 0.05$). Positive correlation was also observed between WBC and LDH concentration at 21 °C in the first (r = 0.73, $p < 0.05$), twenty-fourth (r = 0.64, $p < 0.05$), and forty-eighth hour after exercise (r = 0.75, $p < 0.05$). A correlation was also found between WBC and IL-6 concentration at 21 °C (r = 0.77, $p < 0.05$) in the first hour, and between WBC and HSP concentration at 24 h after physical exercise (r = 0.80, $p < 0.05$). After exercise at 31 °C, such a relationship was found between WBC and Mb concentration (r = 0.67, $p < 0.05$) as well as between WBC and LDH activity (r = 0.66, $p < 0.05$) in the first hour after physical exercise.

The reduction in body mass caused by dehydration during physical exercise in both temperatures led to a decrease in plasma volume ($\Delta$PV%). These changes were not statistically significant, both in the measurements between temperatures of 21 °C and 31 °C and in the measurements separately at temperatures of 21 °C and 31 °C. It is worth noting that athletes performing physical work at 31 °C, where the combined exogenous and endogenous thermal stimuli were more pronounced, depleted the body's water reserves to a

greater extent. In the first hour after the end of physical exercise at 21 °C, the decrease in plasma volume after a series of pulsatile exercises was −4.83%, and at 31 °C, it was at the level of −6.75% (Δ 1.92%).

*3.3. Changes in Selected Markers of Muscle Damage (LDH, Mb), Cytokines (IL-1β, IL-6), and Heat Shock Proteins (HSP70) after Pulsatile Exercise at 21 °C and 31 °C*

Myoglobin is a known biomarker that gives information about muscle tissue damage. In the athletes, its resting concentration at an ambient temperature of 21 °C was 32.86 ng·mL$^{-1}$, and at 31 °C it was 31.39 ng·mL$^{-1}$. After the first hour of exercise at 21 °C, Mb concentration was 40.9 ng·mL$^{-1}$, and at 31 °C it was 54.51 ng·mL$^{-1}$. These changes were statistically significant ($p < 0.05$).

There were statistically significant differences in Mb concentration between the concentration in the first hour after exercise and its concentration at 24 and 48 h after exercise at both temperatures ($p < 0.05$). No significant differences in myoglobin concentration were found between 21 °C and 31 °C separately in individual measurements. Statistical analysis showed a positive correlation between Mb values and E (r = 0.78, $p < 0.05$) and IL-6 (r = 0.80, $p < 0.05$) in the first hour after exercise at 21 °C. A negative correlation was observed at 24 h after exercise at 21 °C between Mb and hGH (r = −0.67, $p < 0.05$) and β-endorphin (r = −0.74, $p < 0.05$). A positive correlation between Mb and LDH was observed in the first hour after exercise at 31 °C (r = 0.64, $p < 0.05$).

The activity of lactate dehydrogenase (LDH), an enzyme present in skeletal muscle cells that leaks into the bloodstream following mechanical muscle damage, increased by 24.32 U/mL in the first hour after a series of pulsatile exercises at 21 °C, and by 42.75 U/mL at 31 °C. These changes in enzyme activity were not statistically significant, either between measurements at 21 °C and 31 °C or between the two temperature conditions separately. As with changes in myoglobin concentration, it can be concluded that work performed at 31 °C resulted in a greater increase in LDH activity in the first hour after physical activity than at 21 °C. This difference was 67.18 U/mL.

A negative correlation was observed in exercise performed at 31 °C at 1 h after exercise between LDH activity and Mb concentration (r = −0.64, $p < 0.05$).

Mechanical, thermal, and metabolic stimuli initiate an inflammatory process as a result of muscle tissue damage. Activated macrophages and monocytes release cytokines, including interleukins serving as pro-inflammatory factors. In the conducted experiment, no significant changes were observed in IL-1β pg/mL and IL-6 pg/mL concentrations between measurements performed separately at ambient temperatures of 21 °C and 31 °C, as well as between ambient temperatures of 21 °C and 31 °C for measurements taken at rest, 1, 24, and 48 h after exercise. It is noteworthy that the baseline values of IL-1β pg/mL and IL-6 pg/mL were higher in athletes who performed interval exercise sequences in an ambient temperature of 31 °C. A negative correlation was found between IL-6 concentration and LDH activity (r = −0.74, $p < 0.05$) at 24 h after exercise at an ambient temperature of 21 °C.

Physical exertion leading to muscle tissue damage and the development of an inflammatory state, which is mainly stimulated by IL-6, also affects the expression of heat shock proteins (HSPs) that participate in the defense of muscle cells against thermal, exercise, or metabolic stress. These changes, like in the case of IL-6 and IL-1β, were not statistically significant between measurements at rest, 1, 24, and 48 h after exercise at both 21 °C and 31 °C temperatures, as well as between 21 °C and 31 °C temperatures separately in measurements. A positive correlation was found between HSP70 concentrations at 24 h and IL-1β at 48 h after physical exertion at 21 °C (r = 0.71, $p < 0.05$).

## 4. Discussion

Effective training for combat sports athletes, including judo athletes, should focus on enhancing speed–strength, endurance, technical–tactical skills, and volitional qualities [34,35]. Sports performance is also shaped by specialized training programs rooted in

physiological and biochemical research [36–38]. These insights help gauge current levels of endurance and speed–strength, identify intensity zones, and track physiological and biochemical shifts from specialized training. It is vital to adapt exercise intensity and recovery to each athlete's potential. Optimal sports performance arises from harmonizing these elements to reach peak fitness values. However, striving for peak performance can lead to adverse physiological changes, overloading the body and risking over-training [31]. Due to the demanding nature of judo, thorough psychophysical preparation is necessary, warranting research on functional and somatic variables influencing matches. Judo athletes require high speed–strength and strong respiratory/circulatory fitness [37]. While aerobic capacity ($VO_2max$) predicts effective fighting, anaerobic endurance plays a crucial role in judo. Notably, Polish judokas' anaerobic endurance indicators serve as a model. Training should enhance tissue oxygenation mechanisms, aiding regeneration and thermoregulation, vital for effective fighting [39,40].

This study's practical significance lies in aiding training staff and combat sports athletes in program construction. The anaerobic capacity among the studied judokas was notable, surpassing Polish representatives [41] and aligning with international standards [42]. Their aerobic endurance ($VO_2max$) was $43.23 \pm 7.79$ mL·kg$^{-1}$, lower than elite athletes [43,44]. Judo competitions may occur in thermoneutral or elevated temperatures. Understanding physiological responses helps design optimal training programs, hydration strategies, and aids recovery [45].

In the context of studying the impact of physical exertion on the immune system, varying perspectives exist. General consensus suggests that intense exercise leads to transient immune system changes, while moderate exercise enhances immune function [46]. Athletes might experience greater immunosuppression than those with moderate physical activity. The interaction between exercise and the immune system could be linked to metabolic, hormonal, or physiological shifts. Athletes engaged in exercise under higher temperatures experience more pronounced immune and endocrine responses compared to room temperature [47]. The extent of this response depends on factors like exercise intensity, duration, and environmental conditions. For instance, post-marathon studies showed increased white blood cell (WBC) counts [48], and ultramarathon runners displayed elevated neutrophil counts and pro-inflammatory cytokine concentrations [49].

In this research, significant increases in WBC counts were observed an hour after exercise. Following pulsating exercises at 21 °C, WBCs rose by 155%, and at 31 °C, by 191.8%. Correlations were found between WBC increase and myoglobin (Mb), lactate dehydrogenase (LDH), and interleukin-6 (IL-6) concentrations [50]. Similar patterns were noted by Paulsen et al. [51]. Muscle damage resulting from intense exercise, particularly with eccentric contractions or altered training, is common among athletes [52]. These damages affect the cell membrane, causing increased enzyme levels in the blood [52]. Exercise-induced micro-injuries manifest as muscle pain, reduced strength, and the increased release of muscle proteins (Mb) and enzymes (CK, LDH) into the bloodstream. LDH and Mb are reliable markers of muscle damage and stress from exercise [53]. Studies indicate that changes in these markers might peak within 24 h [54,55]. The type, intensity, and duration of exercise affect the marker levels [56]. In this study, myoglobin concentration significantly increased post-exercise at both temperatures. Changes in Mb correlated with IL-6 levels [50]. Lactate dehydrogenase (LDH) activity increased by 28.2% at 21 °C and 31.66% at 31 °C post-exercise [50]. Monitoring muscle damage markers helps athletes and coaches tailor training loads, safeguarding against unfavorable muscle changes [52].

When exercise is performed in higher temperatures, the inflammatory process might be connected to blood redistribution [57]. Cytokines like IL-6 and IL-1β are released by immune cells, affecting thermoregulatory centers and potentially leading to hyperthermia [58,59].

Scientific findings [60] reveal that cytokine release, including IL-6 and IL-1β, is influenced by exercise type (duration, intensity, muscle engagement). For exercise exceeding 2 h, IL-6 increased 120-fold and IL-1β by 3-fold. Shorter exercises saw a 6-fold IL-6 increase,

while IL-1β remained relatively stable. Eccentric and concentric exercises induced IL-6 mRNA production, suggesting its production is not solely due to muscle damage, possibly linking it to muscle glycogen levels or glucose production. Some theories connect IL-6 to muscle pain post-exercise [61].

IL-6 plays a central role in the body's response to exercise, affecting fatigue perception and concentration. It is released during exercise and can rise significantly during infections [62]. The IL-6 increase might be linked to muscle cell damage, but it is not closely correlated with effector structure damage biomarkers (LDH, CK, Mb). IL-6 also influences muscle mass growth through satellite cell division [63]. IL-6 concentrations change based on exercise type, duration, muscle volume, and thermal stimuli [62]. Exercise-induced temperature elevation can influence IL-6 changes [62]. Antioxidants like vitamins C and E might limit the action of RFT in reducing muscle damage [64].

Training specialization can lead to decreased IL-6 levels due to receptor expression and heightened cell sensitivity [65]. Lower physical activity increases IL-6, while training reduces resting IL-6 levels [62]. In the study, IL-6 concentration increased after exercise in both ambient temperatures. At 21 °C, it increased by 113.5% in the first hour and 61.22% after 24 h, while at 31 °C, it rose by 97.9% and 5.36%, respectively [53].

IL-1β, produced by macrophages and monocytes, triggers an inflammatory response and affects muscle damage. Early studies mistakenly identified IL-1β as the main pro-inflammatory factor; however, it is IL-6 that holds that role [62]. IL-1β's concentration can remain elevated for days post-exercise. It plays a role in skeletal muscle reconstruction and oxidative stress response [66]. This study did not find significant changes in IL-1β concentration due to anaerobic pulsatile exercises [53].

Heat shock proteins (HSPs) have anti-inflammatory effects and protect muscles from damage. They are affected by exercise, thermal stimuli, and stressors [20]. HSP70 concentration decreased by 40% an hour after exercise at 21 °C, and increased by 53.3% after 24 h. At 31 °C, it increased by 7.29% in the first hour and 51.04% in 24 h after exercise [53]. HSPs are influenced by training and temperature variations [25]. Athletes have higher HSP70 levels [25].

Differences in anaerobic power between upper and lower limbs can influence judo training. Higher ambient temperatures affect physiological and biochemical responses, impacting performance [53]. Evaluating muscle damage markers in judokas helps assess damage levels, regeneration effectiveness, and training adjustments. It also aids in understanding muscle damage and recovery processes [53]. Comparing results between novice and experienced judokas could provide insights into the impact of effort intensity on muscle damage [53].

*Limitation of the Study*

The main limitation was the inability to extend the study with additional measurements after 48 h due to the fact that the study was conducted on a group of elite athletes from the international level during the competition season, which prevented them from staying too long in the research center due to their busy schedule. Inflammatory and interstitial damage markers were measured within 48 h after the test.

## 5. Conclusions

The aim of the conducted observations was to examine the physiological and biochemical reactions of judo athletes to pulsating anaerobic physical exertion in various ambient temperatures. Based on my previous research, it is known that exogenous thermal factors affect muscle energy efficiency, the course of fatigue processes, and the size of micro- and macro-damage to muscle cells. So far, we have not come across scientific information on the degree of differentiation of the physiological response of judo athletes to interval anaerobic exercises in a warm environment. Obtaining this knowledge should enable training staff to optimize training programs and technologies while maintaining ongoing control of the adaptive changes of the body.

During physical exertion, there was an intensification of the immune system response, as evidenced by a significant post-exertion increase in the number of white blood cells. After physical exertion at 31 °C, the increase in white blood cells was 155% and was 36% higher than after exertion at room temperature. The number of leukocytes (WBC) one hour after exertion positively correlated with serum myoglobin concentration and dehydrogenase activity. Exercise-induced muscle damage initiated an inflammatory state, resulting in the release of pro-inflammatory interleukins into the blood.

Based on the obtained research results, it can be concluded that physical exertion in different thermal environmental conditions activates the immune system response to a varying degree, with the direction of changes dependent on the external thermal factor. In order to increase the body's tolerance to thermal and exertion stress, it is advisable for professional judo athletes to conduct training in different ambient thermal conditions.

Based on the obtained research results, the following conclusions have been drawn:

1. Physical efforts in the applied environmental temperatures caused an increased reactivity of the immune system, manifested by a significant increase in the number of leukocytes.
2. Pulsating efforts in two different environmental temperatures contributed to the formation of muscle cell damage, as evidenced by a significant increase in myoglobin concentration. This conclusion has practical significance, as it creates the possibility of programming and optimizing anaerobic training loads.
3. Pro-inflammatory interleukins and lactate dehydrogenase are less sensitive indicators of muscle damage in intense physical efforts.

**Author Contributions:** Conceptualization, T.P., P.M.K. and K.W.; methodology, T.P., Ł.R., E.S.-K. and B.V.; software, T.P.; validation, T.P.; formal analysis, T.P., S.W. and P.K.; investigation, T.P.; resources, T.P. and S.W.; data curation, T.P.; writing—original draft preparation, T.P., P.M.K., E.S.-K., M.M., T.A. and Ł.R.; writing—review and editing, T.P., P.M.K., Ł.T., Ł.R., E.S.-K., P.K., T.A. and Ł.R.; visualization, T.P. and B.V.; supervision, T.P., M.M., T.A. and Ł.T.; project administration, T.P., Ł.T. and K.W.; funding acquisition, T.P. All authors have read and agreed to the published version of the manuscript.

**Funding:** This research received no external funding.

**Institutional Review Board Statement:** The experiment was approved by the Bioethics Committee at the Regional Medical Chamber in Kraków (No. 102/KBL/OIL/2011, 14 December 2011).

**Informed Consent Statement:** Informed consent was obtained from all subjects involved in the study.

**Data Availability Statement:** The data presented in this study are available on request from the corresponding author.

**Conflicts of Interest:** The authors declare no conflict of interest.

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
