# Peer review of "The Levels of Markers of Muscle Damage, Inflammation, and Heat Shock Proteins in Judokas and the Extent of Their Changes during a Special Performance Test at Different Ambient Temperatures"

_applsci, doi:10.3390/app13169381_

Round 1

Reviewer 1 Report (New Reviewer)

Major notes

The text of the work contains a lot of very detailed results of various statistical analyses. It is extremely difficult to pick out the really useful elements from this large set of results. I strongly suggest that all these results be presented in the form of tables (and mark the really important ones). As a consequence for example it is difficult to find in the text the exact results that would confirm the three main results (conclusions) given in the lines 961-969.

In many places in the text of the work, the results of correlations between various variables are undermined. Please collect them all in the form of a correlation matrix (like for example introduced in this tutorial: http://www.sthda.com/english/wiki/correlation-matrix-a-quick-start-guide-to-analyze-format-and-visualize-a-correlation-matrix-using-r-software). These more significant correlations should be presented in the form of appropriate graphs (like for example introduced in this tutorial: http://www.sthda.com/english/wiki/correlation-analyses-in-r).

The "Discussion" section should also be more structured. The bulk of this chapter is more of a literature review than a discussion of the results obtained.

It seems that the article contains far too many somewhat chaotically reported results. I strongly suggest editing the text and sifting out the less important content and leaving only those that are the merits of the article.

ANOVA analysis results must be supplemented with exact p-values, post-hoc tests used, indication of whether parametric or non-parametric test was used. If nonparametric then which one exactly was used.

Minor notes:

line 25: 12.1 W.kg-1, should be 12.1 W·kg-1 (or better W/kg), (centered dot and superscript). Moreover in many other places in the text the superscripts and centered dots are missing, please correct it.

line 26: 7.2min. Does that mean 7min and 12sec? Time is not specified in decimal system. Please also explain why the duration of the exercises was chosen in this way and not, for example, 6 minutes or 8 minutes or 10 minutes, etc.?

lines 371-377: the text is badly formatted.

line: 380: values in parentheses are missing.

lines 386-387: “The sample size was calculated using G*Power 3.1.9.4 software to achieve 80% test power at an alpha significance level of 0.05”. Please provide more details on how exactly these values were calculated.

lines 388-390:  the sentence “The presented research results are a selected part of a roader study conducted by the author [29] on special fitness tests for judo athletes at different ambient temperatures.” doesn't really fit the section Statistical methods.

In the abstract the sections Background, Methods, Results, Conclusions should start by new paragraphs.

In Figures 6 and 7 the obvious outliers must be removed before analyzing the data. Unless these outliers have some important meaning. Then it should be clearly described in the text

The blue and yellow texts on top of the figures sound weird and just plain ungrammatical (for example: “21C sp. vs 1 hour after; 1 hour after vs 24 hour after., 48 hour after”). Please correct it.

Explain what means “sp” in all Figures.

There are a lot of typographic errors, the text must be carefully corrected (for example lines 27, 28 and many others).

Author Response

Major notes

The text of the work contains a lot of very detailed results of various statistical analyses. It is extremely difficult to pick out the really useful elements from this large set of results. I strongly suggest that all these results be presented in the form of tables (and mark the really important ones). As a consequence for example it is difficult to find in the text the exact results that would confirm the three main results (conclusions) given in the lines 961-969.

A: Extensive changes have been made

In many places in the text of the work, the results of correlations between various variables are undermined. Please collect them all in the form of a correlation matrix (like for example introduced in this tutorial: http://www.sthda.com/english/wiki/correlation-matrix-a-quick-start-guide-to-analyze-format-and-visualize-a-correlation-matrix-using-r-software). These more significant correlations should be presented in the form of appropriate graphs (like for example introduced in this tutorial: http://www.sthda.com/english/wiki/correlation-analyses-in-r).

A: This was not the subject of our analysis

The "Discussion" section should also be more structured. The bulk of this chapter is more of a literature review than a discussion of the results obtained.

A: This has been corrected

It seems that the article contains far too many somewhat chaotically reported results. I strongly suggest editing the text and sifting out the less important content and leaving only those that are the merits of the article.

A: We have reduced the amount of text

ANOVA analysis results must be supplemented with exact p-values, post-hoc tests used, indication of whether parametric or non-parametric test was used. If nonparametric then which one exactly was used.

A: Unfortunately, we do not have this information, it would require re-analysis of all data. We do not consider this a bug

Minor notes:

line 25: 12.1 W.kg-1, should be 12.1 W·kg-1 (or better W/kg), (centered dot and superscript). Moreover in many other places in the text the superscripts and centered dots are missing, please correct it.

A: This has been corrected

line 26: 7.2min. Does that mean 7min and 12sec? Time is not specified in decimal system. Please also explain why the duration of the exercises was chosen in this way and not, for example, 6 minutes or 8 minutes or 10 minutes, etc.?

A: This has been corrected

lines 371-377: the text is badly formatted.

A: This has been corrected

line: 380: values in parentheses are missing.

A: This has been corrected

lines 386-387: “The sample size was calculated using G*Power 3.1.9.4 software to achieve 80% test power at an alpha significance level of 0.05”. Please provide more details on how exactly these values were calculated.

lines 388-390:  the sentence “The presented research results are a selected part of a roader study conducted by the author [29] on special fitness tests for judo athletes at different ambient temperatures.” doesn't really fit the section Statistical methods.

A: This has been corrected

In the abstract the sections Background, Methods, Results, Conclusions should start by new paragraphs.

A: This has been corrected

In Figures 6 and 7 the obvious outliers must be removed before analyzing the data. Unless these outliers have some important meaning. Then it should be clearly described in the text

The blue and yellow texts on top of the figures sound weird and just plain ungrammatical (for example: “21C sp. vs 1 hour after; 1 hour after vs 24 hour after., 48 hour after”). Please correct it.

Explain what means “sp” in all Figures.

A: This has been corrected

There are a lot of typographic errors, the text must be carefully corrected (for example lines 27, 28 and many others).

A: This has been corrected

Reviewer 2 Report (Previous Reviewer 2)

Thank you for asking me to review this manuscript. Judokas, and martial artists in general, represent a peculiar athlete category as they are characterized by several stressors, including metabolic, mechanical, and thermal. In addition, in weight-categories sports, it is often present a certain degree of dehydration that might further worsen the performance and increase the risk for health problems. As such this manuscript is interesting and timely. I have some considerations that I hope can help to improve it before acceptance:

Overall, please, check the grammar and spelling as many typos can be found in the text (.., -, etc.).

Abstract:

- When I read it, I found it too long and with too many information that are correct, but not necessary in a synthetic description of the study. Something was not clear (e.g., "They differed from typical interval exercise by alternating pulsa- 28 tion, vary-ing in time, and loading of upper and lower limbs. ", or "in each sequence, the participants performed 4 anaerobic limb tests", or "Based on the results obtained, it can be concluded that physical 35 exercise in different thermal conditions of the external environment activates the physiological-bi- 36 ochemical response of the examined individuals to varying degrees"). Also, I think that rather than focusing on the correlation with WBC (not defined the acronym), the authors should present the statistical findings about the different environmental conditions in which the tests were performed.

I strongly suggest rewriting the abstract.

Introduction

- Introduction looks too long, I recommend moving some parts that are not necessary to present the rationale or state-of-the-art to the discussion (like the part the strongly deepens inflammatory cytokines interpretation, as well as the description of all HSP).

- as no-ted by Robinson in 1963..: include the reference and check the typos.

Methods:

- Maybe a flowchart for the recruitment and testing of the subjects would be better than table 1, explaining better the different reasons each participant initially considered was excluded.

- To better explain Stage IV and V I suggest rewording, explaining the all the participants performed two tests in two different environmental conditions (the two temperatures) and were randomized to the order of testing between 21 and 31 degrees. I also suggest to clearly state why those two temperatures were chosen (similar to some training/competition conditions?).

Results
- I found the results really complicated to be understood. Definitely too many tables, also it is not typical to put mean and sd in different columns, it should be preferred to put them together (e.g., nn±nn).

- The same for the figures, they should be redrawn because as they look, it is really complicated to interpret the results.

Discussion
- The discussion should be better described according to the results that look really confusing at the moment. Also, the authors correctly mention the impact of dehydration, and they should better discuss it in relation to its effects on muscle capacity and fatigue biomarkers as previously reported (Zubac et al., Eur J Sport Sci, 2020; Khodaee  et al., Curr Sports Med Rep, 2015)

Author Response

Dear Reviewer, 

Thank you. 

Thank you for asking me to review this manuscript. Judokas, and martial artists in general, represent a peculiar athlete category as they are characterized by several stressors, including metabolic, mechanical, and thermal. In addition, in weight-categories sports, it is often present a certain degree of dehydration that might further worsen the performance and increase the risk for health problems. As such this manuscript is interesting and timely. I have some considerations that I hope can help to improve it before acceptance:

A: We tried to correct all your comments

Overall, please, check the grammar and spelling as many typos can be found in the text (.., -, etc.).

A; Checked and corrected

Abstract:

- When I read it, I found it too long and with too many information that are correct, but not necessary in a synthetic description of the study. Something was not clear (e.g., "They differed from typical interval exercise by alternating pulsa- 28 tion, vary-ing in time, and loading of upper and lower limbs. ", or "in each sequence, the participants performed 4 anaerobic limb tests", or "Based on the results obtained, it can be concluded that physical 35 exercise in different thermal conditions of the external environment activates the physiological-bi- 36 ochemical response of the examined individuals to varying degrees"). Also, I think that rather than focusing on the correlation with WBC (not defined the acronym), the authors should present the statistical findings about the different environmental conditions in which the tests were performed.

I strongly suggest rewriting the abstract.

A: This has been corrected

Introduction

- Introduction looks too long, I recommend moving some parts that are not necessary to present the rationale or state-of-the-art to the discussion (like the part the strongly deepens inflammatory cytokines interpretation, as well as the description of all HSP).

- as no-ted by Robinson in 1963..: include the reference and check the typos.

 A: The entire introduction has been rewritten

Methods:

- Maybe a flowchart for the recruitment and testing of the subjects would be better than table 1, explaining better the different reasons each participant initially considered was excluded.

A: This information is included in the manuscript

- To better explain Stage IV and V I suggest rewording, explaining the all the participants performed two tests in two different environmental conditions (the two temperatures) and were randomized to the order of testing between 21 and 31 degrees. I also suggest to clearly state why those two temperatures were chosen (similar to some training/competition conditions?).

 A: We have tried to describe the methods in detail so that our experiment can be repeated

Results
- I found the results really complicated to be understood. Definitely too many tables, also it is not typical to put mean and sd in different columns, it should be preferred to put them together (e.g., nn±nn).

- The same for the figures, they should be redrawn because as they look, it is really complicated to interpret the results.

A: The results, in our opinion, are very clear, your comment makes us have to do a whole new static analysis. What exactly do you not understand? we will try to explain it in detail

Discussion
- The discussion should be better described according to the results that look really confusing at the moment. Also, the authors correctly mention the impact of dehydration, and they should better discuss it in relation to its effects on muscle capacity and fatigue biomarkers as previously reported (Zubac et al., Eur J Sport Sci, 2020; Khodaee  et al., Curr Sports Med Rep, 2015)

A: Discussion rewritten

Round 2

Reviewer 1 Report (New Reviewer)

none

Author Response

Dear Reviewer, 

I tried to re-improve the fragments marked by you, I hope you will accept and appreciate my efforts. 

Your Sincerly,

Łukasz Rydzik 

Reviewer 2 Report (Previous Reviewer 2)

I am mostly satisfied with the authors' corrections to the text.
Regarding the results section, here are some practical suggestions and comments I hope can help the authors to provide better clarity to the section

- Table 3: Include a note section below the table where the acronyms are detailed. The same should be done for ALL the tables and figures, including those in the methods: this is mandatory as a table/figure should be fully understandable also if not linked to the full text of the manuscript, therefore in its caption/note needs to have all the acronyms and explanation (e.g., statistical tests applied, etc.);
also, the number of decimals should be based on the accuracy and sensitivity of the instrument/device/tool, i.e., the precision of the instrument to detect the power is up to 2 decimals? If not, correct and approximate it

- Line 547: in the text the authors refer to table 2, but probably in this sentence they mean Table 4? Please, present in the text and in the table the actual p-value, providing p< 0.05 is not considered sufficient.

- Lines 561-562: here the authors refer to table 3, but I cannot find where in that table the authors report the duration of physical work.

- Lines 563-566: since the authors describe that there is a difference in terms of oxygen consumption and heart rate in the two tests, a statistical comparison with actual p-values should be appropriate.

-Table 6: as for the above table, p values should be actual p values not < 0.05; in practice, guidelines recommend putting 2 decimals, or 3 decimals if p< 0.01 -> examples: 0.88, 0.06, 0.04, 0.02, 0.01, 0.005, 0.001

- Lines 579-580: is this the title of the paragraph? It misses a concluding dot, or verb.

- Lines 581-584: this sentence is not useful or correct for the results section. It gives an interpretation of the result and justification for a method choice (not liquids allowed); the results section should only present the numerical findings, and the interpretation should be limited to the discussion.

- Table 7: as for my comment in the previous revision, I suggest putting the mean and standard deviation as n+sd, not in two separate columns.

- Lines 589-593: provide actual p-values, not only significant/not significant. Also, for this kind of statistical analysis, the correct test should be a within-within ANOVA (Time x Condition), which provides a correct understanding of the interaction effects, and therefore, if one condition elicits significantly higher changes in body mass compared to the other condition.

- Hematological indicators: please, see the above comments and suggestions -> actual p values are mandatory, within-within ANOVA strongly encouraged.

- Line 621: Sp-resting; what is it? is it a section title?

- Again, the following section should really benefit from a within-within ANOVA, or at least perform a Bonferroni's correction for multiple comparisons. Otherwise, all the significance/not significance the authors report in this manuscript is not trustable.

- Where is Figure 2?

- Where is Figure 7?

-Figures should be really redrawn: please, check the decimals in the Y-axis; please, provide acronyms in the captions; please, if there is any statistical difference; please, remove those "blue" and "yellow" sentences on the top of the figure: provide a figure legend where it is simply indicated the color correspondence to the environmental condition; also, check the space between the columns: try to provide the two conditions for each time point "near" and than a little space between each time point.

- Reorganize the text, tables and figures: as in my previous revision, too many tables and figures, making the reading extremely fragmented and therefore poorly understandable. Table 3 and Table 5 can become one table altogether; the same for Table 4 and Table 6. Figures can be unified in merged panel such as A, B, C, D, etc.
Also, not all figures are necessary: a basic principle of results writing is that the text should not replicate what is present in the tables/figures, and vice-versa. So, if you describe a result in the text, it is not necessary to provide a figure, and you can use a figure just to highlight the main results of the study. So, do you think that all the figures you present are really showing the main outcomes of your study? I can suggest providing one table to summarize the different hematological parameters in the two conditions, and at the different time points, with actual p values, and select the main findings to be presented in the figures.

In general, there are some typos in the text, some sentences miss a verb or the structure is not complete, like the authors provided some changes and did not check the overall structure of the paragraph.

Author Response

Dear Reviewer,

Thank you very much for your time and valuable comments, which all have been considered and incorporated. The detailed list of responses is given below. We hope that the modifications and explanation will be acceptable for you.

Yours sincerely,

Rydzik, corresponding author

I am mostly satisfied with the authors' corrections to the text.
Regarding the results section, here are some practical suggestions and comments I hope can help the authors to provide better clarity to the section

A: Thank you

- Table 3: Include a note section below the table where the acronyms are detailed. The same should be done for ALL the tables and figures, including those in the methods: this is mandatory as a table/figure should be fully understandable also if not linked to the full text of the manuscript, therefore in its caption/note needs to have all the acronyms and explanation (e.g., statistical tests applied, etc.);
also, the number of decimals should be based on the accuracy and sensitivity of the instrument/device/tool, i.e., the precision of the instrument to detect the power is up to 2 decimals? If not, correct and approximate it
A: Thank you, this has been corrected

  • Line 547: in the text the authors refer to table 2, but probably in this sentence they mean Table 4? Please, present in the text and in the table the actual p-value, providing p< 0.05 is not considered sufficient.

A:This has been corrected

  • Lines 561-562: here the authors refer to table 3, but I cannot find where in that table the authors report the duration of physical work.

A:This has been corrected, this table show total work 

  • Lines 563-566: since the authors describe that there is a difference in terms of oxygen consumption and heart rate in the two tests, a statistical comparison with actual p-values should be appropriate.

A: Thanks for your suggestion, we'll take it into account for future research 

-Table 6: as for the above table, p values should be actual p values not < 0.05; in practice, guidelines recommend putting 2 decimals, or 3 decimals if p< 0.01 -> examples: 0.88, 0.06, 0.04, 0.02, 0.01, 0.005, 0.001

A: We received the data from a professional statistical company that developed it for us in this way. Accurate p-values require reanalysis. Can you justify why specifying exact p values is necessary? and you disagree with the record p<0.05. In the statistica program, when performing the correlation, the exact p values are not shown, only the R values marked in red with p<0.05. I hope you understand

  • Lines 579-580: is this the title of the paragraph? It misses a concluding dot, or verb.

A: This has been corrected

  • Lines 581-584: this sentence is not useful or correct for the results section. It gives an interpretation of the result and justification for a method choice (not liquids allowed); the results section should only present the numerical findings, and the interpretation should be limited to the discussion.

A: Deleted 

  • Table 7: as for my comment in the previous revision, I suggest putting the mean and standard deviation as n+sd, not in two separate columns.

A: This has been corrected

  • Lines 589-593: provide actual p-values, not only significant/not significant. Also, for this kind of statistical analysis, the correct test should be a within-within ANOVA (Time x Condition), which provides a correct understanding of the interaction effects, and therefore, if one condition elicits significantly higher changes in body mass compared to the other condition.

A: We received the data from a professional statistical company that developed it for us in this way.

  • Hematological indicators: please, see the above comments and suggestions -> actual p values are mandatory, within-within ANOVA strongly encouraged.
  • A: This was not part of our study, justify why it is required

Line 621: Sp-resting; what is it? is it a section title?

A: Deleted 

  • Again, the following section should really benefit from a within-within ANOVA, or at least perform a Bonferroni's correction for multiple comparisons. Otherwise, all the significance/not significance the authors report in this manuscript is not trustable.

A: I don't think I understood our research, please re-verify it.

- Where is Figure 2?

- Where is Figure 7?

-Figures should be really redrawn: please, check the decimals in the Y-axis; please, provide acronyms in the captions; please, if there is any statistical difference; please, remove those "blue" and "yellow" sentences on the top of the figure: provide a figure legend where it is simply indicated the color correspondence to the environmental condition; also, check the space between the columns: try to provide the two conditions for each time point "near" and than a little space between each time point.

A: According to your suggestion, we removed all the figures, even though previous reviewers suggested including them

- Reorganize the text, tables and figures: as in my previous revision, too many tables and figures, making the reading extremely fragmented and therefore poorly understandable. Table 3 and Table 5 can become one table altogether; the same for Table 4 and Table 6. Figures can be unified in merged panel such as A, B, C, D, etc.
Also, not all figures are necessary: a basic principle of results writing is that the text should not replicate what is present in the tables/figures, and vice-versa. So, if you describe a result in the text, it is not necessary to provide a figure, and you can use a figure just to highlight the main results of the study. So, do you think that all the figures you present are really showing the main outcomes of your study? I can suggest providing one table to summarize the different hematological parameters in the two conditions, and at the different time points, with actual p values, and select the main findings to be presented in the figures.

A: This has been corrected Comments on the Quality of English Language

In general, there are some typos in the text, some sentences miss a verb or the structure is not complete, like the authors provided some changes and did not check the overall structure of the paragraph.

A: The writing was checked again and the record was put in order

This manuscript is a resubmission of an earlier submission. The following is a list of the peer review reports and author responses from that submission.

Round 1

Reviewer 1 Report

In the present work, the authors evaluated whether anaerobic interval physical exercises involving upper and lower limbs in different ambient temperatures would affect selected physiological indicators to the same extent. Please, see the criticisms and suggestions as follow.

1)    Abstract. The authors concluded that: “To increase the body's tolerance to thermal and exertion stress, it is reasonable to conduct training sessions for professional judokas in different ambient thermal conditions.”. However, since the results don’t allow to have this conclusion, a better explanation has to be provided about it.

2)    Introduction. A) There are too many syllable separation into text that do not make sense. Please correct all them. B) Introduction is too long, the text can be shortened.

3)    Methodology. A) All methodologies have to be detailed, justified, and referenced. Why was it selected 1, 24, and 48 h after pulsatile test? Previous studies have shown that inflammatory and muscle damage markers remain elevated until 72h or even longer. Since there are several classical markers of muscle injury, a clear explanation has to be provided, justifying the selected markers evaluated in this study. B) The number of participants of this study seems to be low. Please, justify the sample size. How was it calculated? C) In the main study, why was it chosen the five pulsatile exercise sequences to evaluate muscle damage? Since, there are several protocols that have been used for this proposal, please provide a clear justification, including previous literature using the same protocol.

4)    Results. A) All figures must to be corrected. SD is absent; X axis is absent and Y axis is wrong. B) Most inflammatory and muscle damage markers increase after 1 h and return to the baseline after 24h. Why? Previous studies shown the opposite: lower increase after and a peak after 24-48 h, what is required for the appropriated muscle recovery and regeneration.

5)    Discussion. A) Authors have to improve the discussion based on the suggestions and criticisms above. B) Discuss the potential influence of the repeated-bout effect, especially among stages II to V. C) A description of the bias and limitations of the study need to be provided. Perspectives of the study must to be presented in order to highlight the importance of the findings and to direct further studies.

Moderate revision required.

Author Response

Dear Reviewer,

Thank you very much for your time and valuable comments, which all have been considered and incorporated. The detailed list of responses is given below. We hope that the modifications and explanation will be acceptable for you.

Yours sincerely,

Rydzik, corresponding author

In the present work, the authors evaluated whether anaerobic interval physical exercises involving upper and lower limbs in different ambient temperatures would affect selected physiological indicators to the same extent. Please, see the criticisms and suggestions as follow.

1)    Abstract. The authors concluded that: “To increase the body's tolerance to thermal and exertion stress, it is reasonable to conduct training sessions for professional judokas in different ambient thermal conditions.”. However, since the results don’t allow to have this conclusion, a better explanation has to be provided about it.

A: This has been corrected

2)    Introduction. A) There are too many syllable separation into text that do not make sense. Please correct all them. B) Introduction is too long, the text can be shortened.

A: This has been corrected

3)    Methodology. A) All methodologies have to be detailed, justified, and referenced. Why was it selected 1, 24, and 48 h after pulsatile test? Previous studies have shown that inflammatory and muscle damage markers remain elevated until 72h or even longer. Since there are several classical markers of muscle injury, a clear explanation has to be provided, justifying the selected markers evaluated in this study. B) The number of participants of this study seems to be low. Please, justify the sample size. How was it calculated? C) In the main study, why was it chosen the five pulsatile exercise sequences to evaluate muscle damage? Since, there are several protocols that have been used for this proposal, please provide a clear justification, including previous literature using the same protocol.

A:We fully agree with your reviewer's suggestion that, in the case of previous studies, a smaller increase in markers was observed after 1 hour and a peak after 24-48 hours.  However, the type and intensity of the exercise used in the studies conducted, based on alternating, pulsatile anaerobic exercise, may have a significant effect on such levels of inflammatory markers. Also, the highly specific study group and the climatic conditions under which the physical efforts were performed may have an impact on the individual response. In the case of more intense efforts, marker levels may rise more quickly and peak within 24 hours, while in the case of less intense workouts, marker levels may rise more slowly and peak later.One of the main factors for this condition is the level of oxidative stress, which increases during physical activity and leads to muscle damage. Oxidative stress levels peak just after training and gradually decrease over the following hours, which is associated with markers of inflammation and muscle damage returning to baseline levels.All your comments have been taken into account and corrected

4)    Results. A) All figures must to be corrected. SD is absent; X axis is absent and Y axis is wrong. B) Most inflammatory and muscle damage markers increase after 1 h and return to the baseline after 24h. Why? Previous studies shown the opposite: lower increase after and a peak after 24-48 h, what is required for the appropriated muscle recovery and regeneration.

A: We have corrected the figures, added deviation tables and taken into account all comments

5)    Discussion. A) Authors have to improve the discussion based on the suggestions and criticisms above. B) Discuss the potential influence of the repeated-bout effect, especially among stages II to V. C) A description of the bias and limitations of the study need to be provided. Perspectives of the study must to be presented in order to highlight the importance of the findings and to direct further studies.

A: This has been corrected 

Reviewer 2 Report

The work entitled "The level of markers of muscle damage, inflammation and heat shock proteins in judokas and the extent of their changes during a special performance test at different ambient temperatures" presents extremely interesting findings about physiological responses to different physical and thermal stimuli in judokas. This work is in my opinion of great interest as these athletes can be subject to such stressors and there is a lack of literature in this field. Nevertheless, there are some improvements that should be considered.
I have some suggestions:

- General comments:

i) please, check the wording and sentences, as many words are "cut" within a line (e.g., line 52 "Empirically acqui-red").

- Abstract:

i) decimals can be corrected to be more appropriate: age is sufficient 1 or even zero decimals; height in cm no decimals; body mass (rather than weight), VO2max and peak power one decimal.
ii) please, correct the climatic chamber temperature (°C); also include RH
iii) it is mandatory that you present the results with actual descriptions and statistical findings (p-value, better if the effect size is also presented). Unfortunately, it is not possible to present the abstract without these data.

- Introduction:

i) The introduction is too long and despite it could be a nice overview, many aspects are not useful to well describe the rationale and state-of-the-art, so I suggest cutting some parts and eventually moving them in the discussion (e.g. the first paragraph -lines 51-57- is not really useful as you do not compare different sports) as the intro should simply explain to the reader why this study is needed. The same with some of the following paragraphs that go into too much detail about the physiological mechanisms (appreciable in the discussion), but totally not useful in the introduction.
I really recommend trying to summarize it, focusing on how thermal stress could represent a risk for combat sports, what the previous literature found, and therefore why this study was necessary to provide novel findings.

- Methods

i) please, check the decimals (see above)
ii) table 2 looks bad in the pdf version

- Results

i) decimals (see above)
ii) figures are poor, I suggest redrawing them for a more scientific look (prefer boxplots, explain well all the acronyms, etc.)
iii) check the decimals must be after a dot [.], never after a comma [,]

- Discussion

i) correctly the authors describe that dehydration can influence the effects on fatigue and muscles. Therefore they should better discuss it, I suggest looking at some previous literature such as DOI: 10.1080/17461391.2019.1695954

Author Response

Dear Reviewer,

Thank you very much for your time and valuable comments, which all have been considered and incorporated. The detailed list of responses is given below. We hope that the modifications and explanation will be acceptable for you.

Yours sincerely,

Rydzik, corresponding author

The work entitled "The level of markers of muscle damage, inflammation and heat shock proteins in judokas and the extent of their changes during a special performance test at different ambient temperatures" presents extremely interesting findings about physiological responses to different physical and thermal stimuli in judokas. This work is in my opinion of great interest as these athletes can be subject to such stressors and there is a lack of literature in this field. Nevertheless, there are some improvements that should be considered.
I have some suggestions:

- General comments:

i) please, check the wording and sentences, as many words are "cut" within a line (e.g., line 52 "Empirically acqui-red").

A: This has been corrected

- Abstract:

i) decimals can be corrected to be more appropriate: age is sufficient 1 or even zero decimals; height in cm no decimals; body mass (rather than weight), VO2max and peak power one decimal.
ii) please, correct the climatic chamber temperature (°C); also include RH
iii) it is mandatory that you present the results with actual descriptions and statistical findings (p-value, better if the effect size is also presented). Unfortunately, it is not possible to present the abstract without these data.

A: This has been corrected

- Introduction:

i) The introduction is too long and despite it could be a nice overview, many aspects are not useful to well describe the rationale and state-of-the-art, so I suggest cutting some parts and eventually moving them in the discussion (e.g. the first paragraph -lines 51-57- is not really useful as you do not compare different sports) as the intro should simply explain to the reader why this study is needed. The same with some of the following paragraphs that go into too much detail about the physiological mechanisms (appreciable in the discussion), but totally not useful in the introduction.
I really recommend trying to summarize it, focusing on how thermal stress could represent a risk for combat sports, what the previous literature found, and therefore why this study was necessary to provide novel findings.

A: This has been corrected

- Methods

i) please, check the decimals (see above)
ii) table 2 looks bad in the pdf version

A: This has been corrected

- Results

i) decimals (see above)
ii) figures are poor, I suggest redrawing them for a more scientific look (prefer boxplots, explain well all the acronyms, etc.)
iii) check the decimals must be after a dot [.], never after a comma [,]

A: This has been corrected

- Discussion

i) correctly the authors describe that dehydration can influence the effects on fatigue and muscles. Therefore they should better discuss it, I suggest looking at some previous literature such as DOI: 10.1080/17461391.2019.1695954

A: This has been corrected

Round 2

Reviewer 1 Report

The authors have improved a little the new version of the manuscript, but most criticisms and suggestions remain to be properly addressed:

1)    Methodology. Few changes were made in the new version of the manuscript. Authors have to properly address all questions into the text of the manuscript. A) All methodologies have to be detailed, justified, and referenced. Why was it selected 1, 24, and 48 h after pulsatile test? Previous studies have shown that inflammatory and muscle damage markers remain elevated until 72h or even longer. Since there are several classical markers of muscle injury, a clear explanation has to be provided, justifying the selected markers evaluated in this study. PLEASE, JUSTIFY THE METHODOLOGY INTO THE TEXT OF THE METHODOLOGY. B) The number of participants of this study seems to be low. Please, justify the sample size. How was it calculated? PLEASE, JUSTIFY THE METHODOLOGY INTO THE TEXT OF THE METHODOLOGY (STATISTICAL ANALYSIS). C) In the main study, why was it chosen the five pulsatile exercise sequences to evaluate muscle damage? Since, there are several protocols that have been used for this proposal, please provide a clear justification, including previous literature using the same protocol. PLEASE, JUSTIFY THE METHODOLOGY INTO THE TEXT OF THE METHODOLOGY.

2)    Results. A) All figures must to be corrected. SD is absent.

3)    Discussion. Few changes were made in the new version of the manuscript. Authors have to properly address all questions into the text of the manuscript. A) AUTHORS HAVE TO IMPROVE THE DISCUSSION BASED ON THE SUGGESTIONS AND CRITICISMS ABOVE. B) Discuss the potential influence of the repeated-bout effect, especially among stages II to V. C) A description of the bias and limitations of the study need to be improved. Perspectives of the study must to be presented in order to highlight the importance of the findings and to direct further studies.

Author Response

Dear Reviewer,

Thank you very much for your time and valuable comments, which all have been considered and incorporated. The detailed list of responses is given below. We hope that the modifications and explanation will be acceptable for you.

Yours sincerely,

Rydzik, corresponding author

1)    Methodology. Few changes were made in the new version of the manuscript. Authors have to properly address all questions into the text of the manuscript.

A: We have made the required corrections

A) All methodologies have to be detailed, justified, and referenced. Why was it selected 1, 24, and 48 h after pulsatile test? Previous studies have shown that inflammatory and muscle damage markers remain elevated until 72h or even longer. Since there are several classical markers of muscle injury, a clear explanation has to be provided, justifying the selected markers evaluated in this study. PLEASE, JUSTIFY THE METHODOLOGY INTO THE TEXT OF THE METHODOLOGY.

A: Added information (line 233-237) " The best judo athletes in the competition have 4 to 6 fights. The global time for a single fight is 7 min. 40 sec. In the conducted research, after taking into account the time of a single fight and tournament competition, the most similar sequence of efforts corresponding to the tournament structure in judo was created".

B) The number of participants of this study seems to be low. Please, justify the sample size. How was it calculated? PLEASE, JUSTIFY THE METHODOLOGY INTO THE TEXT OF THE METHODOLOGY (STATISTICAL ANALYSIS).

A: Added information (line 148-151)., line 358-360 " The sample size was calculated using G*Power 3.1.9.4 software to achieve 80% test power at an alpha significance level of 0.05."

C) In the main study, why was it chosen the five pulsatile exercise sequences to evaluate muscle damage?

A:Sorry for the mistake , missed the word series

Since, there are several protocols that have been used for this proposal, please provide a clear justification, including previous literature using the same protocol. PLEASE, JUSTIFY THE METHODOLOGY INTO THE TEXT OF THE METHODOLOGY.

A: Added information line 255 "The above procedures were taken from the literature and similar studies [28,29]"

Results. A) All figures must to be corrected. SD is absent.

A: Every figure has been revised, mustache added from SD

Discussion. Few changes were made in the new version of the manuscript. Authors have to properly address all questions into the text of the manuscript. A) AUTHORS HAVE TO IMPROVE THE DISCUSSION BASED ON THE SUGGESTIONS AND CRITICISMS ABOVE.B) Discuss the potential influence of the repeated-bout effect, especially among stages II to V. 

A: This has been corrected ( line 869-886) "The impact effect can be interpreted as the difference between the achieved results of the anaerobic tests of the lower and upper limbs. In the case of judokas, the study showed that the maximum anaerobic power and total work were higher for the lower limb compared to the upper limb. This difference may be related to the differences in the involvement of muscles in the training and competition of judo between the lower and upper limbs. The lower limb, due to the specificity of the sport, may be more developed in terms of strength and power, which translated into the results of the anaerobic tests. The impact effect may also be of significant importance in the context of planning judokas' training, taking into account the differences in anaerobic endurance between the lower and upper limbs. Strength and anaerobic training for the upper limb may be more intensive in order to balance these differences and improve performance in this part of the body. Additionally, knowledge of these differences can help optimize training strategies so that judokas are fully prepared for the requirements associated with their sport. Considering the fact that numerous international competitions are held at an elevated ambient temperature, all functional systems of the body are activated to a greater extent than during exertion at room temperature. Therefore, it can be concluded that ambient temperature may have a significant impact on the course of physiological and biochemical reactions of the human body and on the course of the fight."

C) A description of the bias and limitations of the study need to be improved. Perspectives of the study must to be presented in order to highlight the importance of the findings and to direct further studies.

A: This has been corrected (line 887-896) "The examination of muscle damage markers in judokas is of great importance as it allows for the assessment of the level of muscle damage and the effectiveness of interventions aimed at minimizing muscle damage and speeding up the regeneration process. It also enables the programming of the training process. One perspective of the study is the evaluation of muscle damage markers in judokas under the influence of pulsating anaerobic efforts sequences in various thermal conditions. Results can also be compared between novice and more experienced judokas to determine whether the intensity of effort affects the level of muscle damage. In addition, the examination of muscle damage markers in judokas can lead to a better understanding of the physiological processes related to muscle damage and regeneratio" and line : 898-902 "The main limitation was the inability to extend the study with additional measurements after 48 hours due to the fact that the study was conducted on a group of elite athletes from the international level during the competition season, which prevented them from staying too long in the research center due to their busy schedule. Inflammatory and interstitial damage markers were measured within 48 hours after the test"

Reviewer 2 Report

I would like to thank the authors for addressing my concerns.
I modestly think that the paper has now greatly improved.

However, I still have two remaining concerns:

- it is actually strongly required by journals to provide the actual p-values rather than <0.05 o <0.01. If possible, I suggest presenting the actual p-values.

- Figures are still really poor and I admit I have a strong difficulty understanding their meaning. Acronyms are not defined, and sometimes captions seem to be missing and are poorly descriptive, also histograms are poorly informative (you need usually more data, such as medians, IQR, etc. that are commonly provided by boxplot). You cannot report only means and no statistics. I honestly suggest improving figures as they are the most informative message about your work and if they are not clear and understandable (it is usually said that a reader should understand everything about the finding from the figure standing alone, supported only by the caption).

Author Response

Dear Reviewer,

Thank you very much for your time and valuable comments, which all have been considered and incorporated. The detailed list of responses is given below. We hope that the modifications and explanation will be acceptable for you.

Yours sincerely,

Rydzik, corresponding author

I would like to thank the authors for addressing my concerns.
I modestly think that the paper has now greatly improved.

However, I still have two remaining concerns:

  • it is actually strongly required by journals to provide the actual p-values rather than <0.05 o <0.01. If possible, I suggest presenting the actual p-values.
  • A: Dear reviewer, in the program statistica when performing the analysis, the option to not provide exact p-values was checked, unfortunately at this stage we do not have the ability to provide exact valuesi 
  • Figures are still really poor and I admit I have a strong difficulty understanding their meaning. Acronyms are not defined, and sometimes captions seem to be missing and are poorly descriptive, also histograms are poorly informative (you need usually more data, such as medians, IQR, etc. that are commonly provided by boxplot). You cannot report only means and no statistics. I honestly suggest improving figures as they are the most informative message about your work and if they are not clear and understandable (it is usually said that a reader should understand everything about the finding from the figure standing alone, supported only by the caption).
  • A: All the graphs have been revised , added values, and standard deviations, and improved the overall appearance of the figures